# Effect of hydrogen on the integrity of aluminium–oxide interface at elevated temperatures

Meng Li[1], De-Gang Xie[1], Evan Ma[1,2], Ju Li[1,3], Xi-Xiang Zhang[4] & Zhi-Wei Shan[1]

Hydrogen can facilitate the detachment of protective oxide layer off metals and alloys. The degradation is usually exacerbated at elevated temperatures in many industrial applications; however, its origin remains poorly understood. Here by heating hydrogenated aluminium inside an environmental transmission electron microscope, we show that hydrogen exposure of just a few minutes can greatly degrade the high temperature integrity of metal–oxide interface. Moreover, there exists a critical temperature of $\sim 150\,°C$, above which the growth of cavities at the metal–oxide interface reverses to shrinkage, followed by the formation of a few giant cavities. Vacancy supersaturation, activation of a long-range diffusion pathway along the detached interface and the dissociation of hydrogen-vacancy complexes are critical factors affecting this behaviour. These results enrich the understanding of hydrogen-induced interfacial failure at elevated temperatures.

[1] Center for Advancing Materials Performance from the Nanoscale (CAMP-Nano), State Key Laboratory for Mechanical Behavior of Materials, Xi'an Jiaotong University, Xi'an 710049, China. [2] Department of Materials Science and Engineering, Johns Hopkins University, Baltimore, Maryland 21218, USA. [3] Department of Nuclear Science and Engineering and Department of Materials Science and Engineering, Massachusetts Institute of Technology, Cambridge, Massachusetts 02139, USA. [4] Division of Physical Science and Engineering, King Abdullah University of Science & Technology, Thuwal 23955-6900, Saudi Arabia. Correspondence and requests for materials should be addressed to J.L. (email: liju@mit.edu) or to X.-X.Z. (email: xixiang.zhang@kaust.edu.sa) or to Z.-W.S. (email: zwshan@mail.xjtu.edu.cn).

Metals and alloys take up hydrogen during their processing and service, when exposed to humid atmosphere, hydrogen-containing gases or aqueous solutions[1,2]. Hydrogen in these materials can modify several aspects of defect behaviour[3–5] that are closely related to failure modes such as hydrogen embrittlement[6–8], cavitation/blistering[9] and interface failure[10–12], and thus greatly undermine the material reliability. Although protective films such as aluminium oxide and chromium oxide are widely adopted as environmental barriers, their protection against hydrogen is incomplete; they may even retard hydrogen effusion and facilitate defect formation, such as dislocation loops, micropores and blisters[12–17]. Such hydrogen-induced damage is known to be exacerbated at a few hundred degree Celsius, including typical service temperatures under hydrogen-containing environment in gas turbines, power plants, petrochemical factories and solar sails[18–23].

Previous works have shown that in hydrogenated metals, blisters usually grow with increasing temperature[14,23–25]. However, occasionally they could also shrink or disappear[26,27]. The accelerated blister growth is often interpreted as simply due to the rise of internal gas pressure in the hydrogen-filled cavities[24,25], while the reason for shrinkage is still elusive. However, there are other important factors that can instigate and mediate blistering but have not been taken into consideration fully. First of all, it was recently revealed that surface-diffusion-driven metal cavitation process plays a critical role in the nucleation and growth of blisters at room temperature[28]. Thus, besides gas pressure, metal diffusion must also play a key role in hydrogen-related interface failure. Second, superabundant vacancies are regularly observed in hydrogenated metals and alloys regardless of the hydrogenation method[29,30]. By reducing the formation energy of vacancies, hydrogen can stabilize vacancy and vacancy clusters[3,4,6–8,31]. Atomistic simulations show that hydrogen and vacancy tend to form hydrogen-vacancy complexes, with a rather high binding energy[32–35] and migration energy[36,37]. Hence, the hydrogenated vacancies are much more stable[36] and diffuse much slower than bare vacancies, lowering their probability to reach sinks and making it easier to accumulate to superabundant concentrations upon temperature change, plastic deformation and/or radiation that drive the system out of equilibrium. Although it has been speculated that these vacancies are likely to affect the cavitation or blistering process[31–34,38], the underlying mechanism is not fully understood. Careful experiments are therefore needed to reveal the interplay of these factors to better model the mechanisms responsible for the development of blisters or cavities at high temperatures.

In this work, we probe into the mechanistic details by performing real-time dynamical observations at the metal–oxide interface. Using state-of-the-art environmental transmission electron microscopy (ETEM) and a home-made ultra-stable heating stage, the microstructural evolution for hydrogenated and hydrogen-free aluminium samples was directly monitored and compared *in situ* during heating with quantitative temperature control. Single crystalline cylindrical aluminium pillars were studied to have a good edge-on view of the metal–oxide interface. Both hydrogenated and hydrogen-free pillars were simultaneously heated to 200 °C under constant heating rate ($0.3$ °C s$^{-1}$) in vacuum ($<10^{-4}$ Pa). Our home-made microelectromechanical system (MEMS) heating chip has minimal thermal drift in all three dimensions including the e-beam direction, allowing ultra-stable observations of the evolution of these cavities throughout the heating process with one-to-one correlation to the real-time temperature. Our observations show that hydrogen exposure can greatly

undermine the integrity of the metal–oxide interface at elevated temperature. The loss of integrity is accompanied by the formation of giant cavities only in hydrogenated pillars. By careful analysis of cavity volume, it is further confirmed that the hydrogen-induced superabundant vacancies played a critical role in the cavity evolution process. Our TEM experiments are conducive to studying the effects of such vacancies, because the illuminating 300 keV electron beam radiation can displace Al atoms off lattice sites and generates excess vacancies. These vacancies mimic, and can be considered representative of, those originating from other out-of-equilibrium situations encountered in service, such as electrochemical/chemical/plasma hydrogenation, thermal quenching, plastic deformation and so on, facilitating the detection of the role played by the superabundant vacancies.

## Results

**Hydrogen induced formation of giant cavity.** Two separate groups of pillar samples were fabricated via focused ion beam (FIB) micromachining, each on an aluminium plate attached to the hotplate of the MEMS heating chip using FIB lift-out process (Fig. 1 and Supplementary Note 1). The nominal crystal orientation and size of these pillars were identical. One group of samples was first hydrogenated inside the ETEM with 2 Pa H$_2$ at room temperature (20 °C) using the method introduced in our previous work[28]. The hydrogen source was cut off when very small Wulff construction-shaped cavities ($R \sim 5$ nm, referred to as

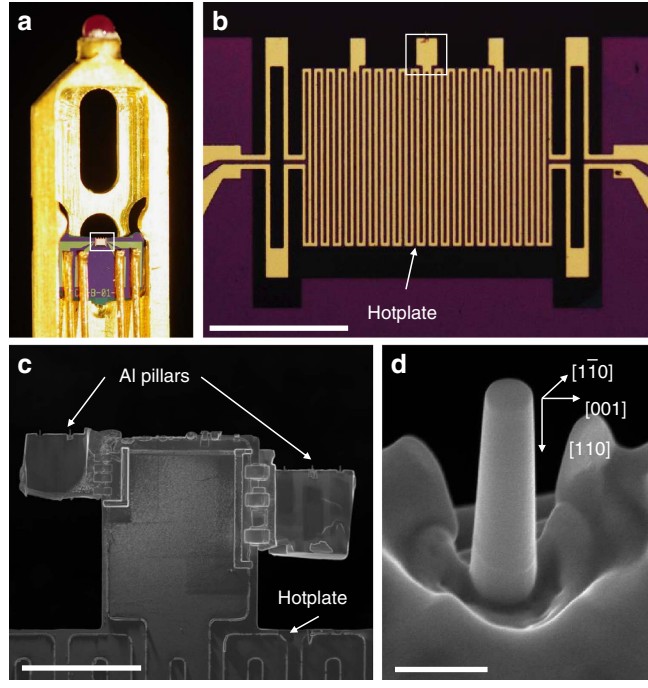

**Figure 1 | Experimental setup and sample information.** (**a**) Optical image of the front end of the TEM specimen holder with a home-made MEMS heating chip mounted inside. (**b**) Enlarged view of the area indicated by white rectangle in (**a**) showing the heating/sensing traces and sample mounting bars in the hotplate of the MEMS heating chip. (**c**) SEM image of the sample-mounting area as outlined by a white square in (**b**) with two lift-out aluminium plates attached at both sides. The Al pillars are directly fabricated on the Al plates using FIB. (**d**) SEM image of a typical FIBed pillar viewed from 45°. The pillar is ~250 nm in diameter with an axial direction of [110]. The pillar is viewed from the [1$\bar{1}$0] direction in TEM. Scale bar: (**b**) 200 μm, (**c**) 30 μm, (**d**) 500 nm.

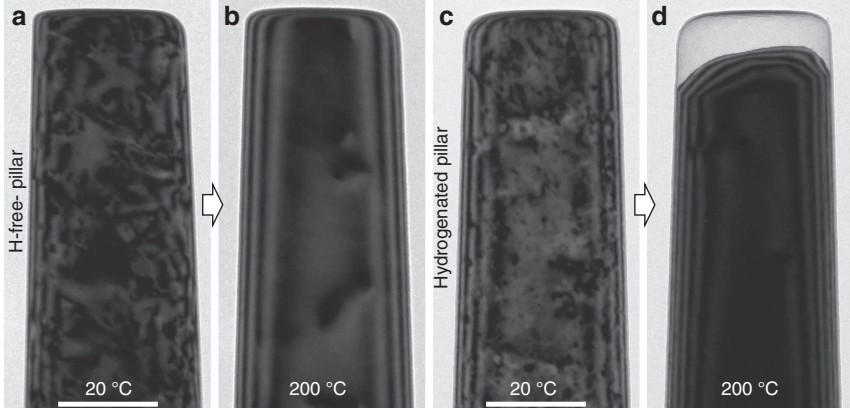

**Figure 2 | Comparison of heating effect on hydrogen-free and hydrogenated pillars.** (**a,b**) The morphological change of a hydrogen-free pillar. At room temperature (**a**), the interior of the pillar shows high density of dislocations that are annealed out after heating up to 200 °C (**b**). After heating, a clean interior and smooth thickness contours are observed. (**c,d**) The morphological change of a hydrogenated pillar. Besides annealing, a giant cavity formed at the top of the pillar. Scale bar, 200 nm.

proto-cavities hereafter) became visible at the metal–oxide interface while the overall geometrical profile of the pillars remained unchanged. The other group of the pillars was exposed to electron beam at the same intensity and for a comparable time in vacuum as a control reference to rule out the effect from electron-beam irradiation only. After this step, hydrogenated and hydrogen-free pillars were simultaneously heated to 200 °C under constant heating rate (0.3 °C s$^{-1}$) in vacuum (<10$^{-4}$ Pa).

After heating to 200 °C, giant cavities with detached oxide shell and naked-metal surface were observed to form in hydrogenated pillars but not in hydrogen-free pillars (see Fig. 2). Before heating, both the hydrogen-free pillar (Fig. 2a) and the hydrogenated pillar (Fig. 2c) contained a high density of long dislocation lines and dislocation loops. After the samples were heated to 200 °C, nearly all of these dislocations were annealed out from both samples, as evidenced by the nearly perfect thickness contours inside the pillars (Fig. 2b,d). However, within the hydrogenated pillar, a giant cavity of ∼300 nm in size was observed to form via retreat of the naked-metal surface at the front end of the pillar, leaving behind a neat hollow oxide shell (Fig. 2d).

**Cavity evolution process**. Figure 3 and the Supplementary Movie show the development of proto-cavities in the hydrogenated pillar with increasing temperature (see Supplementary Fig. 1 for snapshots of the Supplementary Movie). Figure 3a–d are the top left side of the pillar at 20, 100, 150 and 200 °C, respectively. At 20 °C, the proto-cavities show a homogenous distribution at the oxide–metal interface, with an average diameter of ∼10 nm. In the ensuing heating-up process, the evolution of these cavities can be divided into three stages. (1) Before reaching 100 °C, some proto-cavities grow by coalescing with neighbouring proto-cavities (Fig. 3a,b and Supplementary Fig. 1b,c) with a mechanism similar to Ostwald ripening. (2) Between 100 and 150 °C, the size of most cavities remains largely unchanged or even slightly decreases, while a few bigger ones continue to increase in size slightly (Fig. 3c and Supplementary Fig. 1d). (3) Between 150 and 200 °C, nearly all proto-cavities decrease in size until they eventually disappear, while one proto-cavity at the free end of the pillar develops into a giant cavity (Fig. 3d and Supplementary Fig. 1e,f). The formation of this giant cavity is also similar to Ostwald ripening, although it happens over a greater distance than that in the proto-cavity growth in stage 1, by absorbing proto-cavities

located within micrometres distance. The size evolution of a proto-cavity (solid square) and the giant cavity (hollow square) versus temperature/time are plotted in Fig. 3e that clearly shows the divergence of size change after a critical temperature ($T_c$) of 150 °C, likely because after reaching $T_c$, the large-scale coalescence of cavities becomes possible via long-range diffusion pathways, following the ripening of proto-cavities before reaching $T_c$, when connections between neighbouring detached areas formed.

One intuitive thought is that the volume of the giant cavity should be equal to the sum of those disappeared small proto-cavities. However, to our surprise, this is not true as will be detailed below. To measure the volume change before and after heating, some pillars were illuminated at only the protruding part during hydrogenation process. After hydrogenation, proto-cavities were only found in the electron beam radiated area (see Supplementary Fig. 2a–c). After heating the sample to 200 °C, only one giant cavity formed at the free end of the pillar (Supplementary Fig. 2d). This experimental design allowed us to estimate the change in total cavity volume before and after heating. We found that those proto-cavities contributed a maximum of ∼1/3 to the volume of the giant cavity (see Supplementary Note 2 for details). In addition, the Supplementary Movie and Supplementary Fig. 1e,f also show that the giant cavity continued to expand at the metal side after 200 °C, when all proto-cavities had disappeared and therefore could no longer feed volume to the giant cavity. Thus, our results indicate the existence of another source of volume.

**Distribution of cavities**. Giant cavities can be found in many sites of the hydrogenated volume, as shown in Fig. 4. Note that the two cavities marked with black arrows in Fig. 4a (left side of panel) are each larger than those along the top surface of the pillar. Also note that after heating up to 200 °C, these cavities coalesced, developing into a giant cavity along the side of the pillar, while the cavities along at all the other sites became refilled (Fig. 4a, right side of panel). By accident, we found the second electron image collected by through-lens detector (TLD) are surprisingly effective in detecting the buried cavities. Figure 4b shows a typical scanning electron microscopy (SEM) image of the hydrogenated-and-heated sample taken with the TLD, in which giant cavities (dark contrast areas) can be found in several places at the plane surfaces and at the pillar top, with one

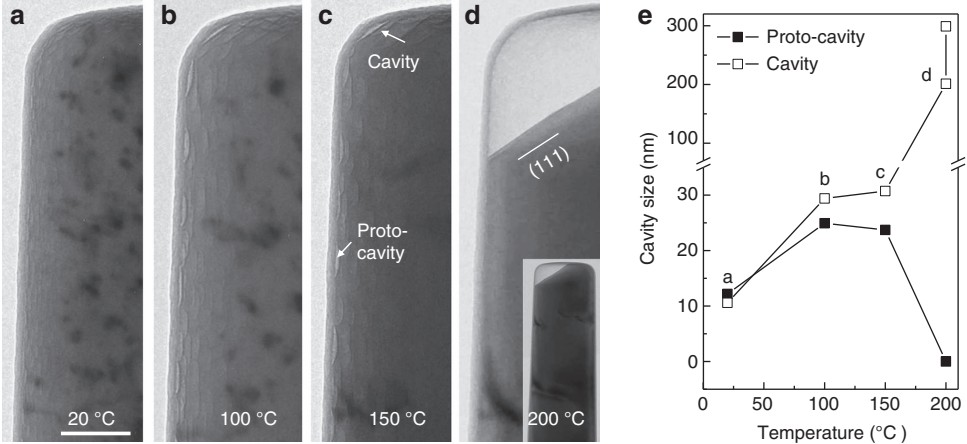

**Figure 3 | Formation of a typical giant cavity at AlOx–Al interface.** See the Supplementary Movie for the dynamic process. (**a**–**d**) Snapshots of the pillar at 20, 100, 150 and 200 °C, respectively. (**e**) Size evolution of the proto-cavities and cavity with temperature. From 20 to 100 °C, the proto-cavities grow, but as the cavity on the corner begins to grow, the proto-cavities decrease in size. Scale bar, 100 nm.

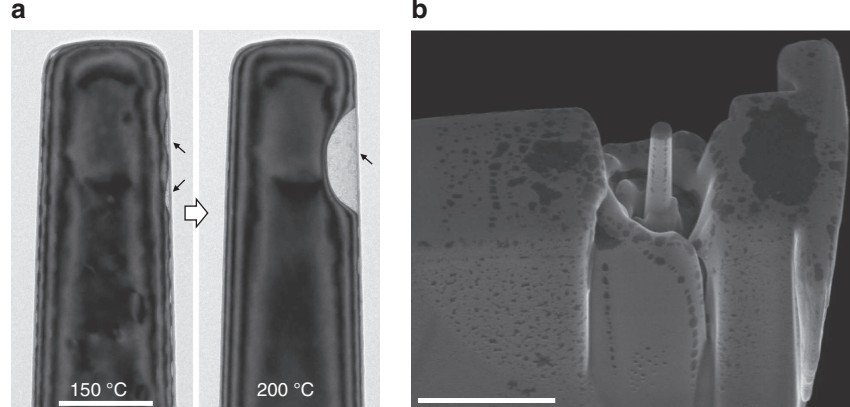

**Figure 4 | Distribution of giant cavities on sample surface.** (**a**) Formation of a giant cavity at the same location as the largest proto-cavities at the side of the pillar, as marked by the arrows. (**b**) SEM image viewed from 45° of a larger sample area after the same hydrogenation and heating treatments. Giant cavities not only formed at the top of the pillar, but also at the plane surface of the plate. Scale bar: (**a**) 200 nm, (**b**) 2 μm.

reaching up to ∼2 μm in size. It is worth noting that using conventional imaging conditions, that is, the Everhart–Thornley detector, these cavities were invisible and showed only a uniform contrast, proving the necessity of using TLD images for cavity detection in industrial applications.

## Discussion

Hydrogenated metals and alloys are prone to having super-abundant vacancies regardless of the hydrogenation method[29,30]. Inside the microscope, the high-energy electron beam radiation and hydrogen are expected to contribute to the formation of superabundant vacancies inside the illuminated volume[39]. If the extra volume in our experiment comes from the vacancy concentration, then their density is estimated to be ∼$10^4$ atomic p.p.m., and this is high but similar to that reported by Buckley *et al.*[30] through plasma charging. The existence of the superabundant vacancies was supported by the careful examination of the TEM images before and after the hydrogenation process that showed that the diameter of the pillar increased slightly (Supplementary Fig. 3), and the increased volume matches well with the extra volume of the giant cavity (see Supplementary Fig. 2 and Supplementary Note 2 for details).

At room temperature, these vacancies are stable and contribute little to interfacial cavitation, because the incorporated hydrogen atoms can stabilize the vacancies with a high binding energy to form hydrogen-vacancy complexes ($VaH_n$) that in turn slow down their diffusion and limit their elimination at grain boundaries or free surfaces[32–36,40]. However, as the temperature increases, these hydrogen-vacancy complexes are expected to be dissociated (due to entropic effects), producing large numbers of freed hydrogen interstitials and bare vacancies: $VaH_n \rightarrow Va + nH$[38]. The destabilizing temperature has been measured by different techniques to be ∼100–200 °C (refs 27,33,41). This encompasses the $T_c \sim 150$ °C observed in our experiments. As shown in the Supplementary Movie and Supplementary Fig. 1b, at room temperature, lots of small black speckles were formed during the e-beam hydrogenation process. These speckles are small interstitial dislocation loops created by radiation damage of the hydrogenated metal[39]. These dislocation loops remained unchanged during the heating process until ∼150 °C, when all these dislocation loops disappeared, followed by the rapid growth of the giant cavity (Supplementary Fig. 1d). Presumably, the disappearance of the interstitial loops is due to recombination with the freed vacancies from the decomposition of hydrogen-vacancy complexes at this critical temperature.

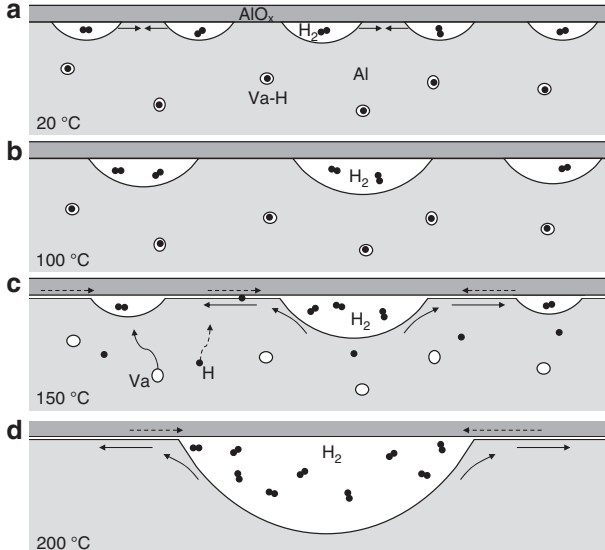

**Figure 5 | Schematic illustration of the formation of giant cavities in hydrogenated metals during heating.** (**a**) At room temperature, the hydrogenated metals have gas-filled proto-cavities under the oxide layer and hydrogenated vacancies in the bulk. (**b**) From 20 to 100 °C, the proto-cavities coalesce to grow bigger. (**c**) From 100 to 150 °C, the hydrogenated vacancies tend to dissociate due to the thermal destabilization. Both interstitial hydrogen and naked vacancies are freed to diffuse outwards. As a result, the metal–oxide interface becomes further weakened, opening up a diffusion highway for surface diffusion and $H_2$ flow under the oxide layer. The diffusion pathways of Al and $H_2$ are marked with solid and dashed arrows, respectively. (**d**) At 200 °C, a giant cavity is formed where the biggest proto-cavities was found.

Therefore, the volume contribution from hydrogenated vacancies can be rationalized by the dissociation of hydrogenated vacancies as well, destabilized by the time-dependent temperature rise. With less or no hydrogen atoms, the vacancy can diffuse faster to segregate at interfaces and giant cavities.

In the early stage of giant cavity formation, the metal surface usually shows low-energy surface facets of {111} (Fig. 3d), indicating a surface diffusion-mediated cavity evolution (see Supplementary Fig. 4 and Supplementary Note 3). At room temperature, surface diffusion mainly occurs locally, but at higher temperatures, surface diffusion becomes effective across longer distances, enabling the coalescence of numerous proto-cavities into a few giant cavities. This long-range surface diffusion at high temperature may be activated by the following two processes: on one hand, the growth and initial coalescence of proto-cavities connect more and more detached interfacial areas (Fig. 3b,c), opening up a long-range fast diffusion pathway. On the other hand, the temperature rise drives more dissolved hydrogen and supersaturated vacancies out of the substrate. When these effusing vacancies are absorbed by the interface, they will break more interfacial bonds and detach more metal–oxide interface that further boosts the interfacial/surface diffusion under the oxide shell. To verify the latter argument, we aged some hydrogenated samples in vacuum at room temperature for 12 h to effuse the hydrogen and therefore reduce the number of hydrogen-vacancy complexes significantly. After ageing, the interfacial proto-cavities remained unchanged and heating thereafter to 200 °C resulted in a similar disappearance of proto-cavities, but no giant cavity form (see Supplementary Fig. 5). What happened instead was the previously detached oxide layer re-bonded with the naked metal to remove the proto-cavities, and

shutting down naked-metal surface diffusion. The dissociation of hydrogen-vacancy complex is thus necessary for opening up the metal–oxide interface to allow naked-metal surface diffusion that enabled the production of Wulff-construction-shaped proto-cavities and their subsequent aggregation into giant cavities.

Hydrogen damage of passivated metal surfaces at high temperatures can be summarized as the four-step process shown in Fig. 5. Step 1, during hydrogenation at room temperature, due to the low solubility of hydrogen in aluminium at room temperature[42], hydrogen mainly segregates at defective traps such as vacancies and the metal–oxide interface. By reducing the ideal work of interface separation[5,43], the confinement of oxide shell on the metal matrix is weakened that further promotes the surface-diffusion-driven cavitation under the metal–oxide interface. Since the segregated hydrogen atoms at interface have chance to meet and recombine, the produced hydrogen molecules will fill into those cavities. The above formation process of gas-filled cavities has been detailed in our previous work[28]. Besides, by trapping hydrogen atoms, vacancies in the bulk are stabilized at $T < 150$ °C and become superabundant (compared with those without hydrogen). Both the proto-cavities and the hydrogenated vacancies play important roles in the ensuing cavitation processes. Step 2, when pillars are heated up to 100 °C, proto-cavities in the material grow up by absorbing neighbouring cavities via Ostwald ripening. Step 3, when the critical temperature is approached, the hydrogenated vacancies dissociate to produce H interstitials and bare vacancies, both of which have much higher mobility than hydrogenated vacancies. The interstitial H atoms and bare vacancies can diffuse quickly into the metal–oxide interface, weakening and even breaking more interfacial bonds, a phenomenon that can be termed 'interfacial hydrogen decohesion'. This creates a diffusion highway along the interface with the exposed naked-metal surface. Step 4, at 200 °C, via this naked-metal surface diffusion, the proto-cavities begin to shrink, with their empty volumes and filling gas quickly sucked away to expand the few giant cavities, and the higher mobility of the naked superabundant vacancies ($\sim 10^4$ atomic p.p.m., or $\sim 1$ at.%) allows them to diffuse to the interface/surface, contributing further to the growth of the giant cavity.

Although these changes proceed vigorously under the surface oxide layer, they do not cause any obvious changes to the exterior surface morphology. Consequently, these cavities may escape routine optical or SEM inspection and can therefore impose a significant threat to the reliability of the material, such as oxide scale spalling off from turbine blades or accelerated corrosion. Revealing these decohesions using TLD in SEM may be a useful measure to guard against unexpected failure. In addition, the time-domain technique we have established here inside TEM with precise temperature and environment control allows quantitative measurement of the population dynamics of gas-defect complexes, naked-metal surface diffusivity as well as their microstructural consequences under non-equilibrium conditions that can arise in service due to temperature changes and environmental and radiation exposure. Finally, the 'interfacial hydrogen decohesion' method might be a way to isolate large areas of few-nm-thick oxide shell as two-dimensional material.

## Methods
**Sample preparation.** Single-crystalline aluminium (99.9995%) disks were cut into $1.5 \times 2$ mm$^2$ rectangular plates and mechanically polished on one surface. Then, $\sim 30 \times 30$ μm$^2$ lamellas with a thickness of 3 μm were fabricated using FIB (FEI Helios 600, operating at 30 kV) and transferred to the freestanding end of the home-made MEMS heating chip in Fig. 1 using the FIB lift-out process.

The chip was specially designed for samples prepared from bulk material using FIB and can achieve ultra-high spatial stability even during temperature change (see Supplementary Note 1 for details). Cylindrical aluminium pillars with [110] axial direction were fabricated directly on the lamellas after transfer. The milling current used for the last step was as low as $\sim 10$ pA to minimize the FIB damage. Compared with the traditional acute wedge-shaped sample prepared by electrochemical polishing, this cylindrically shaped sample has a controllable surface curvature and is edge-on to the electron beam, hence revealing the microstructural changes at the metal–oxide interface more easily. All pillars had diameters ranging from 200 to 400 nm and an aspect ratio (height/diameter) of $\sim 3$. The as-fabricated pillars had an $\sim 7$ nm thick surface oxide layer.

**In situ TEM experiment.** Hydrogenation and heating were carried out in situ in the differentially pumped Hitachi H-9500 ETEM operating at 300 keV with the home-made MEMS-based ultra-stable heating holder shown in Fig. 1a. The ETEM was evacuated to a base vacuum of $10^{-4}$ Pa.

During the hydrogenation process, 2 Pa ultra-high-purity $H_2$ (99.999%) was introduced to the specimen chamber through a needle valve, measured by a vacuum gauge near the specimen. The specimen was kept at room temperature (20 °C) during the hydrogenation process. Due to the e-beam knock-on effect, some $H_2$ molecules in the e-beam illuminated area dissociated into H atoms and ions and then penetrated the surface oxide layer to reach the aluminium[28]. Using this method, the sample can be hydrogenated very locally even with very low $H_2$ pressure. The degree of hydrogenation can be tuned by adjusting the electron-beam intensity and the illumination time. To rule out FIB effect on the results, both annealed and as-fabricated samples were hydrogenated and heated and the cavity evolution processes were compared (for details, see Supplementary Note 4 and Supplementary Figs 6 and 7).

The heating process was performed in vacuum ($<3 \times 10^{-4}$ Pa) using the heating stage with a constant heating rate ($0.3$ °C s$^{-1}$). The electron beam direction was close to [110] direction of all pillars. Microstructural changes of the pillars during the heating process were recorded using the Gatan 832 CCD camera at 2 frames per s.

**Data availability.** The data that support the findings of this study are available from the corresponding authors on request.

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

## Acknowledgements

M.L., D.-G.X. and Z.-W.S. acknowledge support from the National Natural Science Foundation of China (51231005 and 51621063) and the International Joint Laboratory for Micro/Nano Manufacturing and Measurement Technologies. J.L. acknowledges support by NSF DMR-1120901 and DMR-1410636. E.M. acknowledges support from US DoE-BES-DMSE under Contract No. DE-FG02-09ER46056. M.L. acknowledges the support from King Abdullah University of Science and Technology (KAUST) during

her stay at KAUST as an exchange student. X.-X.Z. acknowledges the support from King Abdullah University of Science and Technology.

## Author contributions

Z.-W.S. and X.-X.Z. designed the project. M.L. and D.-G.X. conducted the experimental work. M.L., D.-G.-X., Z.-W.S., J.L., E.M. and X.-X.Z. wrote the paper. All authors contributed to discussions of the results.

## Additional information

**Competing financial interests:** The authors declare no competing financial interests.

