## [Peer Review File · Nature Communications]

Reviewers' Comments:

Reviewer #1 (Remarks to the Author):

The experimental results presented in the manuscript and the supplementary material is very convincing as they are based on in-situ observations. However, the interpretation is rather speculative. The interface energies and their changes due to hydrogen segregation and concomitant lowering of the ideal work of interface separation is not discussed, although this is known for a while. Some related recent publications are M.C. Tiegel et al. *Acta Mater.* 115 (2016) 24-34 and R. Kirchheim et al. *Acta Materialia* 99 (2015) 87–98. In addition, any discussion of gaseous hydrogen formed in cavities is missing. This is expected to be the case, although it may be difficult to guess as it requires knowledge of the total amount of hydrogen in the sample. However the solubility in equilibrium with gaseous hydrogen is very small and known.

Major revision is recommended taking the previous considerations into account.

Reviewer #2 (Remarks to the Author):

The authors studied the hydrogen effect protective oxide layer on the surface of the aluminium pillars. They use a novel and experimental methodology based on in situ H charging and heating in an ETEM. The results show that hydrogen enhances the formation of superabundant vacancies as well as small cavities in the interface of the oxide and metal. Later during the heating through de-trapping of the H from the superabundant vacancies the and surface diffusion, the cavities are growing together and become a single giant cavity. The volume of the giant cavity is larger than the initial cavities since there is a contribution from the superabundant vacancies.

The experiments are designed and performed excellently. The results are of high interest for the scientific community. However, there are some small concerns which are better to be addressed before the publication of the manuscript.

In line 42 only H precipitation and segregation is discussed. Since later authors consider superabundant vacancies and H enhanced defect formation. I believe it is better to discuss and review a wider range of H effect on defect formation in metals. Referencing to the defectant theory proposed by Kirchheim (Reducing grain boundary, dislocation line and vacancy formation energies by solute segregation, *Acta Mat* 2007) and experimental works by Takai (Hydrogen thermal desorption relevant to delayed-fracture susceptibility of high-strength steels

MMTA) and Vehoff (Recent developments in the study of hydrogen embrittlement: Hydrogen effect on dislocation nucleation, Acta Mat) in this regards could be very useful. Additionally, a wide range of atomistic simulation both ab initio and molecular dynamics supports these experimental works e.g. (Hydrogen-enhanced dislocation activity and vacancy formation during nanoindentation of nickel).

Line 54 "... while the reason of shrinkage is still in lack" need a reference.

Line 60 As mentioned before there more updated references are available than the original works of Birnbaum which are worth to mention.

Line 73. I believe it is necessary to discuss the potential FIB damage and Ga implantation effect on the results. The behavior of non H-charged samples are convincing but still there could be a synergic effect of Ga in H charged sample which is not possible to exclude. However, the FIBing parameters can be provided and estimate of the potential damage from the FIB can be discussed.

Line 161 instead of "evaporating" a more common word in the community "De-trapping" can be used.

Line 215 "Hydrogen de-passivation" term can be a misleading term since it is not "de-passivation" which is the electrochemical process of removal of the passive film at cathodic polarization. For me what happening is the typical "hydrogen blistering".

Reviewers' comments:

Reviewer #1 (Remarks to the Author):

The experimental results presented in the manuscript and the supplementary material is very convincing as they are based on in-situ observations.

Reply: We thank the reviewer for considering us results “very convincing”.

However, the interpretation is rather speculative. The **interface energies and their changes due to hydrogen segregation and concomitant lowering of the ideal work of interface separation is not discussed**, although this is known for a while. Some related recent publications are M.C. Tiegel et al. Acta Mater. 115 (2016) 24-34 and R. Kirchheim et al. Acta Materialia 99 (2015) 87–98. In addition, any **discussion of gaseous hydrogen formed in cavities is missing**. This is expected to be the case, although it may be difficult to guess as it requires knowledge of the total amount of hydrogen in the sample. However the solubility in equilibrium with gaseous hydrogen is very small and known.

Major revision is recommended taking the previous considerations into account.

Reply: Thanks for the constructive suggestion and the useful reference papers.

Accordingly, we have updated the discussion of the hydrogen effect on lowering

interface energy and the hydrogen formed in cavities in the discussion part as listed

below:

...Step 1, during hydrogenation at room temperature, due to the low solubility of hydrogen in aluminum at room temperature⁴³, hydrogen mainly segregates at defective traps such as vacancies and the metal/oxide interface. By reducing the ideal work of interface separation^{5,42}, the confinement of oxide shell on the metal matrix is weakened, which further promotes the surface-diffusion-driven cavitation under the metal/oxide interface. Since the segregated hydrogen at interface have chance to meet and recombine, the produced hydrogen molecules will fill into those cavities. The above formation process of gas-filled cavities has been detailed in our previous work²⁸. Besides, by trapping hydrogen at ...

Reviewer #2 (Remarks to the Author):

The authors studied the hydrogen effect protective oxide layer on the surface of the aluminium pillars. They use a novel and experimental methodology based on in situ H charging and heating in an ETEM. The results show that hydrogen enhances the formation of superabundant vacancies as well as small cavities in the interface of the oxide and metal. Later during the heating through de-trapping of the H from the superabundant vacancies the and surface diffusion, the cavities are growing together and become a single giant cavity. The volume of the giant cavity is larger than the

initial cavities since there is a contribution from the superabundant vacancies. The experiments are designed and performed excellently. The results are of high interest for the scientific community. However, there are some small concerns which are better to be addressed before the publication of the manuscript.

Reply: We thank the reviewer for the very nice summary of our work and considering our results “of high interest for the scientific community”.

In line 42 only H precipitation and segregation is discussed. Since later authors consider superabundant vacancies and H enhanced defect formation. I believe it is better to **discuss and review a wider range of H effect on defect formation in metals**. Referencing to the defactant theory proposed by Kirchheim (Reducing grain boundary, dislocation line and vacancy formation energies by solute segregation, Acta Mat 2007) and experimental works by Takai (Hydrogen thermal desorption relevant to delayed-fracture susceptibility of high-strength steels MMTA) and Vehoff (Recent developments in the study of hydrogen embrittlement: Hydrogen effect on dislocation nucleation, Acta Mat) in this regards could be very useful. Additionally, a wide range of atomistic simulation both ab initio and molecular dynamics supports these experimental works e.g. (Hydrogen-enhanced dislocation activity and vacancy formation during nanoindentation of nickel).

Reply: Thanks for the very nice suggestion to review more H effect on defect formation in metals in the introduction and thanks for the suggested papers. We have

added discussion on it in the introduction part and cited the suggested papers as below:

*Metals and alloys pick up hydrogen during their processing and service, when exposed to hydrogen-containing gases, humid atmosphere, and aqueous solutions^{1,2}. Hydrogen in these materials can modify several aspects of defect/microstructural behavior³⁻⁵, which are closely related to failure modes such as hydrogen embrittlement⁶⁻⁸, cavitation/blistering⁹ and interface failure¹⁰⁻¹², and thus greatly undermine the material reliability. Although protective films...
...the hydrogenation method^{29,30}. By reducing the formation energy of vacancies, hydrogen can facilitate vacancy formation^{3,4,6-8,31}. Atomistic simulations also show that hydrogen and vacancy tends to form hydrogen-vacancy complexes, with a rather high binding energy³²⁻³⁵ and migration energy^{36,37}. Hence, the hydrogenated vacancies are much more stable and diffuse much slower than bare vacancies, lowering its probability to meet the sinks and making it easy to accumulate to a superabundant concentration. Although it has been speculated that...*

Line 54 "... while the **reason of shrinkage** is still in lack" need a reference.

Reply: Thanks for pointing out the missing reference. We have added reference to this sentence as below:

...The accelerated blister growth is often interpreted as simply due to the rise of internal gas pressure in the hydrogen-filled cavities^{18,19}, while the reason for

shrinkage is still in lack...

Line 60 As mentioned before there more updated references are available than the original works of Birnbaum which are worth to mention.

Reply: Thanks for the nice suggestion. Birnbaum's work showed the experiment result of superabundant vacancy concentration, while the suggested work from Kirchheim showed the reason for the superabundant vacancy, and Takai, Vehoff and Wen's work showed experimental and simulation evidence for the existence of superabundant vacancy and its effect on hydrogen embrittlement. We have modified this part into:

...Second, superabundant vacancies are prevailing in hydrogenated metals and alloys regardless of the hydrogenation method^{29,30}. By reducing the formation energy of vacancies, hydrogen can facilitate vacancy formation^{3,4,6-8,31}. Atomistic simulations also show that hydrogen and vacancy tends to form hydrogen-vacancy complexes, with a rather high binding energy³²⁻³⁵ and migration energy^{36,37}. Hence, the hydrogenated vacancies are much more stable and diffuse much slower than bare vacancies, lowering its probability to meet the sinks and making it easy to accumulate to a superabundant concentration. Although it has been speculated that these vacancies are likely to affect the cavitation or blistering process^{31-34,38}, the underlying mechanism is still not understood. ...

Line 73. I believe it is necessary to discuss the **potential FIB damage and Ga**

implantation effect on the results. The behavior of non H-charged samples are convincing but still there could be a synergic effect of Ga in H charged sample which is not possible to exclude.

Reply: Thanks for the nice suggestion. FIB damage and Ga implantation effect are two common problems in every FIB fabricated samples.

Firstly, Ga was reported to segregate at grain boundaries in Aluminum. Since no grain boundary is present in our case, the metal/oxide interface may be the location where Ga should segregate to. We performed electron energy loss spectroscopy (EELS) mapping on the interface of many samples, but found no appreciable Ga signal (see the Fig. R1 below). This suggests that Ga segregation plays insignificant role in the hydrogen related interface evolution process in our sample.

Fig. R1| Chemical analysis at the hydrogenated pillar surface with EELS. (a) A pillar severely hydrogenated under E-beam in 2 Pa H₂ for 55 min. (b) EELS spectrum taken underneath the blister in (a) shows clear O-K edges and Al-K edges, but no Ga-L₃ edges, indicating that Ga segregation is insignificant in our

experiment. Scale bar, 200 nm.

Secondly, thermal annealing experiments were performed to clean out the FIB-induced defects and implanted Ga prior to the hydrogenation process. Then the clean samples were subject to exactly the same hydrogenation and heating process as described in the main article. Comparison of results from the as-FIB pillar and the clean pillar were shown below. We also add these information in the **Supplementary Fig. 7**. The detailed description and discussion on the FIB effect are added to **Supplementary Note 4**.

Supplementary Note 4: FIB effect on the experiment result

*In aluminum, the damages caused by FIB fabrication are point defect clusters formed near sample surface and Ga implantation.^{5,6} The point defect clusters are characterized to be interstitial Frank loops induced by irradiation⁶. Both these FIB-induced defects and implanted Ga can be cleaned out by thermal annealing⁷. **Supplementary Fig. 1a-b** shows the results from thermal annealing at 200 °C. It can be seen that after annealing, the initially ‘dirty’ pillar became clean, as evidenced by the smooth thickness contour. Besides, we observed from SEM image that some spherical particles formed on the lamella surface after annealing, which was proved to be Ga by energy dispersive X-ray analysis (**Supplementary Fig. 6**).*

To verify the FIB effect on the giant cavity formation process, annealed samples were hydrogenated and heated up in vacuum with the same experiment

conditions mentioned in the method section. The comparison of results from as-fabricated pillar and well-annealed pillar are shown in **Supplementary Fig. 7**. After heating, giant cavities are formed in both samples, and the volumes of the giant cavities are comparable. This result indicates that **the FIB-induced defects and Ga implantation have negligible effect on our observed cavity evolution.**

Supplementary Figure 6 | Formation of Ga droplets after heating. (a) SEM image after thermal annealing. The implanted Ga segregated out and form tiny particles (marked by white arrows) on sample surface. (b) Energy dispersive X-ray analysis indicate that these particles are Ga. Scale bar, 5 μm .

Supplementary Figure 7 | The giant cavity formation on both as-fabricated and well-annealed aluminum samples. (a)-(c) giant cavity formation process in un-annealed sample. (d)-(f) giant cavity formation process in annealed sample, the FIB damage induced point defect clusters are removed by annealing before hydrogenation and heating. Scale bars, 200 nm.

However, **the FIBing parameters can be provided and estimate of the potential damage from the FIB can be discussed.**

Reply: The detailed FIB parameters and FIB effect are now added to the methods session as below:

...Cylindrical aluminum pillars with [110] axial direction were fabricated directly on the lamellas after transfer. The used milling current for the last step was as low

as ~10 pA to minimize the FIB damage. Compared with the traditional acute wedge-shaped sample prepared by electrochemical polishing, this cylindrically shaped...

*...The degree of hydrogenation can be adjusted by adjusting the electron-beam intensity and the illumination time. To rule out FIB effect on the results, both annealed and as-fabricated samples were hydrogenated and heated and the cavity evolution process were compared. (For details, see **Supplementary Note 4 and Supplementary Fig. 6-9**)...*

Line 161 instead of "evaporating" a more common word in the community "De-trapping" can be used.

Reply: Thanks for the nice suggestion. We have changed the word into "De-trapping" in the main text:

*...the time-dependent temperature rise destabilizes vacancies by **de-trapping** the hydrogen out of the vacancy...*

Line 215 "Hydrogen de-passivation" term can be a misleading term since it is not "de-passivation" which is the electrochemical process of removal of the passive film at cathodic polarization. For me what happening is the typical "hydrogen blistering".

Reply: Thanks for pointing out the misuse of "de-passivation". We have deleted this new term in our article. Here by using "Hydrogen de-passivation", we originally meant that it is like the reverse phenomenon of "oxide passivation", in which the

metal surface got passivated by oxide layer and compact interfacial bonds are formed.

In the “hydrogen de-passivation” phenomenon, the original compact metal-oxide interface got weakened by hydrogen due to the reduced interface energy and interfacial bonds are broken, resulting in fresh metal surface and detached oxide layer.

Since it turned out that this term may cause misunderstanding, we have changed the word into “Hydrogen decohesion”:

...The interstitial H atoms can diffuse quickly into the metal-oxide interface, weakening and even breaking more interfacial bonds, a phenomenon that can be termed as “hydrogen decohesion” ...

Reviewers' Comments:

Reviewer #2 (Remarks to the Author):

The authors have incorporated my comments and suggestions in the revised version, which is now recommended for publication.

Reviewer #3 (Remarks to the Author):

All concerns regarding the publication of the manuscript are addressed and the manuscript is acceptable for publication.